# Adjuvant Chemotherapy for Pathological Node-Positive Disease in Squamous Cell Carcinoma of the Esophagus after Neoadjuvant Chemoradiotherapy Plus Surgery

**DOI:** 10.3390/jpm12081252

**Published:** 2022-07-29

**Authors:** Jing-Wei Lin, Chuan Li, Hui-Ling Yeh, Cheng-Yen Chuang, Chien-Chih Chen

**Affiliations:** 1Department of Radiation Oncology, Taichung Veterans General Hospital, Taichung 407, Taiwan; owell25@gmail.com (J.-W.L.); i5495108@gmail.com (C.L.); bighelenyeh@gmail.com (H.-L.Y.); 2Department of Radiation Oncology, Chiayi Branch, Taichung Veterans General Hospital, Chiayi City 600, Taiwan; 3Division of Thoracic Surgery, Department of Surgery, Taichung Veterans General Hospital, Taichung 407, Taiwan; cychuangtw@gmail.com; 4Ph.D. Program in Translational Medicine, College of Life Science, National Chung-Hsing University, Taichung 402, Taiwan

**Keywords:** adjuvant chemotherapy, esophageal cancer, squamous cell carcinoma

## Abstract

***Purpose:*** The purpose of the present study is to determine the impact on survival using adjuvant chemotherapy on patients with locally advanced esophageal cancer. ***Materials and Methods:*** From 2007 to 2016, we enrolled 127 locally advanced esophageal squamous cell carcinoma patients treated with combined neoadjuvant chemoradiotherapy (nCRT) and surgery. For patients with the pathological residual primary disease (pT+) and/or residual node disease (pN+) after nCRT, adjuvant chemotherapy was also given after consideration of the toxicity of nCRT, patient performance, and/or comorbidity. The regimen of adjuvant chemotherapy was cisplatin 20 mg/m^2^/day and 5-fluorouracil 800 mg/m^2^/day on days 1 through 4 and 22 through 25. The primary endpoint was overall survival (OS). ***Results:*** From a total of 127 patients, 26 of them (20.5%) received adjuvant chemotherapy. In the multivariate analysis, pN+ diseases were independently associated with poor OS (hazard ratio (HR): 4.117, 95% confidence interval (CI): 1.366–12.404; *p* = 0.012). For those with pN+ diseases, their 5-year OS was 36.4% in the follow-up arm compared with 45.8% in the adjuvant chemotherapy arm (*p* = 0.094). ***Conclusions:*** Pathologic node-positive disease is associated with poor OS in locally advanced esophagus cancer patients after combined treatments with nCRT and surgery. Adjuvant chemotherapy appeared to have improved OS in pathologic node-positive diseases.

## 1. Introduction

Esophageal cancer is the 10th most common cancer worldwide in 2020 and is in the top 5 in cancer-related mortality [1]. In Asian countries, squamous cell carcinoma (SCC) constitutes >90% of all esophageal cancers [1]. The 5-year OS rate of patients with esophageal cancer treated with surgery alone is lower than 25%. More than one-fifth of patients have a microscopic positive surgical margin after primary surgery, and 31% of patients have locoregional failure [2]. Therefore, for those potentially resectable cases, neoadjuvant chemoradiotherapy (nCRT) plus operation is their standard treatment [3,4,5,6]. According to the CROSS trials, the common patterns of recurrence were distant failures (20.7%) and combined locoregional and distant failures (10.8%) [7]. Adding systemic therapies to the standard treatment protocol may theoretically reduce distant failures. However, there is no clear evidence supporting the benefits of adjuvant therapies after nCRT plus operation, and previous studies were mainly carried out on patients with adenocarcinoma of the esophagus. In a retrospective study based on the National Cancer Database, adjuvant chemotherapy is associated with a 31% lower risk of death in esophageal adenocarcinoma with pN+ diseases after nCRT plus operation [8]. In a meta-analysis of 10 studies involving various sample sizes of patients, adjuvant therapy groups had significantly lower mortality at both 1-year (risk ratio (RR) 0.52, 95% CI 0.41–0.65) and 5-year follow-up (RR 0.91, 95% CI 0.86–0.96). However, patients with squamous cell carcinoma only comprised 10.3% of the overall cohort. Besides, multivariable analyses in this meta-analysis did not evaluate the interaction between adjuvant chemotherapy and histological subtypes [9]. The purpose of this study is to clarify the survival impact of adjuvant chemotherapy for locally advanced SCC of the esophagus.

## 2. Materials and Methods

### 2.1. Patients

The inclusion criteria were patients with histologically diagnosed SCC of the esophagus, clinical stage II–IVA (T2-4N1-3M0 or T3-4N0M0; non-metastatic node-positive diseases, or primary tumor invasion to adventitia or adjacent structures) according to the classification of the American Joint Committee on Cancer Tumor-Node-Metastasis (TNM), 8th edition [10,11,12,13,14,15]. We enrolled patients receiving nCRT plus surgical operations at Taichung Veterans General Hospital.

The exclusion criteria were those with any of the following: (a) synchronous and/or metachronous history of other malignancies within 5 years; (b) histology other than SCC; (c) distant metastatic disease at initial diagnosis; (d) previous radiotherapy field covering esophageal and/or mediastinum; (e) induction chemotherapy before nCRT; (f) concurrent chemotherapy other than cisplatin and 5-fluorouracil (5-FU); (g) incomplete treatment; and (h) patient’s refusal or unsuitable for surgery.

A total of 127 patients were enrolled in this study from October 2007 to July 2016. All patients had Karnofsky performance scores of 70 or above before treatment. All patients completed the pretreatment staging workup, including the following: comprehensive medical history, clinical physical examination, esophageal tumor biopsy, chest X-ray, chest computed tomography scan (CT), endoscopic ultrasound, bronchoscopy, complete blood cell count, serologic evaluation of liver and renal functions, and positron emission tomography with 2-deoxy-2-[fluorine-18]fluoro-D-glucose integrated with computed tomography (PET/CT). PET/CT scans were performed within two weeks before nCRT, and then 4 to 5 weeks after the completion of nCRT. All patients received standard-of-care (SOC) with nCRT plus operation. After SOC, adjuvant chemotherapy was assigned to the high-risk patients at the joint discretion of oncologists and patients, while considering toxicities of previous treatment, patient performance and/or comorbidities, and pathological residual disease after nCRT.

Our study was approved by the Institutional Review Board of Taichung Veterans General Hospital (CE21111A, date of approval: 27 April 2021) [16]. A written form of informed consent was obtained from all participating patients. All of the patients were treated according to the National Comprehensive Cancer Network (NCCN) guideline of esophageal cancer.

### 2.2. Neoadjuvant Chemoradiotherapy before Surgery

All 127 patients went through CT simulation in a supine position. The following were outlined on the CT images: gross tumor volume (GTV), clinical target volume (CTV), planning target volume (PTV), and organs at risk (OARs). GTV was defined as the gross tumor of the esophagus and enlarged lymph nodes (based on PET/CT and/or CT scans). CTV of the primary tumor was delineated as the gross esophageal tumor plus a radial margin of 1 cm and a longitudinal margin of 5cm. In the case of upper or middle esophagus tumors, CTV of the nodal target delineation was covering the nodal GTV with a 5 mm margin, the mediastinum, and supraclavicular regions. In the case of lower third esophagus tumors, CTV of the nodal target delineation was covering the nodal GTV with a 5 mm margin and the celiac trunk area. PTV was defined as CTV including a 5 mm margin compensating for daily setup errors and internal organ motions. We delivered a total radiation of doses from 50 to 50.4 Gy over a period from 25 to 28 daily fractions, 1.8 to 2.0 Gy per fraction, and 5 fractions per week. The plan of intensity-modulated radiation therapy (IMRT) and the source–axis distance (SAD) technique was applied using a dynamic multi-leaf linear accelerator with 6 MV photon energy.

The concurrent chemotherapy in nCRT was given with cisplatin at 20 mg/m^2^ and fluorouracil at 800 mg/m^2^ for 24 h on day 1 to day 4 (cycle 1), and day 29 to day 32 (cycle 2) during the course of radiotherapy. Before chemotherapy, patients were checked for granulocyte counts, which were >1500 per mL, as well as platelet counts >100,000 per mL, creatinine clearance >50 mL/min, and normal results in liver function tests.

### 2.3. Surgical Operations

The operation was performed within 4 to 6 weeks after the completion of nCRT. The surgical procedure included thoracoscopic esophagectomy, lymph node dissection, and esophagus reconstruction with the gastric tube.

### 2.4. Pathological Analyses

Pathological examinations included histology type, tumor extension, resection margins, and nodal involvement. The treatment response was assessed using the Mandard tumor regression grade (TRG) [17]. TRG was quantitated in five grades: TRG 1 showed an absence of residual cancer and fibrosis extending through the different layers of the esophageal wall; TRG 2 was characterized by the presence of rare residual cancer cells scattered through the fibrosis; TRG 3 was characterized by an increase in the number of residual cancer cells, but fibrosis still predominated; TRG 4 showed residual cancer outgrowing fibrosis; and TRG 5 was characterized by an absence of regressive changes. Major responses included TRG 1 (complete regression) and TRG 2 (rare residual cancer cells). Pathological complete response (pCR) was defined as the absence of residual primary tumor and positive lymph nodes.

### 2.5. Adjuvant Chemotherapy after Surgery

Adjuvant chemotherapy was given within 2 months post-operation as decided at the joint discretion of oncologists and patients while considering toxicities of previous treatment, patient performance and/or comorbidities, and pathological residual disease after nCRT. Adjuvant chemotherapy was planned for two courses of cisplatin 20 mg/m^2^ and fluorouracil 800 mg/m^2^ on day 1 to day 4 and day 22 to day 25.

### 2.6. Follow-Up and Patient Survivals

Patients received the follow-up survey including endoscopic examination, chest CT scan, abdominal sonography, and bone scan every 3–4 months in the first 3 years after the completion of treatment. In subsequent years, restaging was performed every 6 months. At each hospital visit, patients were evaluated for late toxic effects and disease recurrence. Cases of mortality were documented. The severity of toxicity was graded using the National Cancer Institute Common Terminology Criteria for Adverse Events (CTCAE) version 4.0.

### 2.7. Statistical Analyses

The primary endpoint of this study was overall survival (OS), which was measured from the date of biopsy to the date of all-cause death or the last date of follow-up. The Kaplan–Meier method was used to determine survival outcomes, and inter-group differences were assessed with the log-rank test. Multivariate analyses were performed based on the Cox proportional hazards model for estimating each covariate, the hazard ratio, and 95% confidence interval. Covariates included sex, age, histological differentiation (well to moderately differentiated vs. poorly differentiated), site (cervical esophagus, upper/middle/lower thoracic esophagus), clinical T/N classification, pre-treatment tumor length, and pathological T/N classification. The chi-square test or Fisher’s exact test was used to compare categorical variables between the two groups; that is, with and without adjuvant chemotherapy. The independent samples t-test was used to compare the mean age of the patients with adjuvant chemotherapy to the patients without adjuvant chemotherapy. Statistical significance was set at *p* < 0.05. Statistical analyses were all performed using SPSS software, version 25.

## 3. Results

Patients’ characteristics are shown in Table 1. For the entire cohort, the mean age was 54.77 years old. The median nCRT duration was 39 days (interquartile range (IQR), 35.8–40.2 days). Of the 127 patients, 126 of them (99.2%) received the planned two cycles of concurrent chemotherapy, and 125 of 127 patients (98.4%) finished the full dose of radiotherapy. The median time duration between the date of completion of nCRT and the date of operation was 32 days (IQR 29–36 days). After nCRT plus operation, the pCR rate was 44.1%. The median number of dissected lymph nodes was 31 (range: 10 to 93). The median follow-up of the surviving patients was 59.7 months. The 5-year OS of the entire cohort was 59.1%.

Of the 127 patients, 26 of them (20.5%) received adjuvant chemotherapy. Adjuvant chemotherapy tended to be given to those without pathological complete response (*p* < 0.001) or patients with pathological residual node diseases (pN+) (*p* < 0.001). All of these 26 patients completed two cycles of adjuvant chemotherapy. Acute toxicities during adjuvant chemotherapy are shown in Table 2. In the 26 patients of our adjuvant chemotherapy group, the most common grade 3–4 hematologic toxicity was leucopenia (7.7%), and the most common grade 3–4 non-hematologic toxicities were hypokalemia (11.5%) and hyponatremia (11.5%).

The potential prognosticators of overall survival estimated by univariate and multivariate analyses are shown in Table 3. In the univariable analysis, clinical incomplete response, pN+ diseases, non-pCR, pathologically residual tumor size, the ratio of the number of positive lymph nodes to the total number of lymph nodes dissected, and the lower percentage change of the maximum standardized uptake value (SUVmax) before and after nCRT were associated with poor OS. In the multivariate analysis, patients with pN+ diseases were independently associated with poor OS (hazard ratio (HR): 4.117, 95% confidence interval (CI): 1.366–12.404; *p* = 0.012). Subgroup analyses of OS are presented as the forest plot (Figure 1).

Adjuvant chemotherapy did not yield benefits on OS for the entire cohort. The 5-year OS in the follow-up arm was 62.4% compared with 46.2% in the adjuvant chemotherapy arm (*p* = 0.124) (Figure 2). Adjuvant chemotherapy may have better OS in high-risk patients with pN+ disease. For the pN+ subgroup, the 5-year OS in the follow-up arm was 36.4% compared with 45.8% in the adjuvant chemotherapy arm (*p* = 0.094) (Figure 3).

## 4. Discussion

The majority of newly-diagnosed esophageal cancers are presented with locally advanced diseases. For locally advanced disease after nCRT plus operation, more than 30% of patients have experienced distant failure and only 3.3% of patients have isolated loco-regional recurrences [7]. Because distant metastasis is the most common failure pattern and re-irradiation tends to cause more damage on the adjacent normal organs after nCRT plus operation, adjuvant chemotherapy might be appropriate with the possibility of better tumor control and survival outcomes on selected patients with high risks.

Leng et al. [18]. reported that patients with pN+ diseases after nCRT plus operation have poor OS and disease-free survival. Our patients with pN+ diseases tended to have poor survival. Adjuvant chemotherapy may improve OS in patients with pN+ diseases with borderline significance in the subgroup analysis. Pathologic nodal positive diseases may be an adequate criterion for further adjuvant therapy.

For SCC of the esophagus, the number of reports is very limited on adjuvant chemotherapy after nCRT plus operation. In a retrospective study based on the National Cancer Database, adjuvant chemotherapy is associated with a 31% lower risk of death in esophageal adenocarcinoma with pN+ diseases after nCRT plus operation [8]. In a meta-analysis of 10 studies involving various sample sizes of patients, adjuvant therapy groups had significantly lower mortality at both 1-year (risk ratio (RR) 0.52, 95% CI 0.41–0.65) and 5-year follow-up (RR 0.91, 95% CI 0.86–0.96). However, patients with squamous cell carcinoma only comprised 10.3% of the overall cohort. Besides, multivariable analyses in this meta-analysis did not evaluate the interaction between adjuvant chemotherapy and histological subtypes [9]. In the present study, adjuvant chemotherapy tended to improve OS in those with pN+ diseases. For the pN+ subgroup, the 5-year OS in the follow-up arm was 36.4% compared with 45.8% in the adjuvant chemotherapy arm.

Because of severe acute toxicities of nCRT and compromised performance status after major surgery, it is not straightforward to give all patients adjuvant chemotherapy. There is still no randomized trial of adjuvant chemotherapy in addition to locally advanced squamous cell carcinoma of esophagus patients who received nCRT plus operation. The toxicity profile of nCRT was reported previously [3,4,6,19]. The most common grade 3–4 hematologic toxicity in nCRT is leucopenia (ranging from 6% to 48.8%), and the most common grade 3–4 non-hematologic toxicity is nausea/vomiting (ranging from 2% to 11%). During the nCRT course in our hospital, the most common grade 3–4 hematologic toxicity was leucopenia (19.1%), and the most common grade 3–4 non-hematologic toxicity was esophagitis (8.8%). No grade 3–4 nausea/vomiting was noted. Besides, adjuvant chemotherapy was also safe and tolerable. In the 26 patients of our adjuvant chemotherapy group, the most common grade 3–4 hematologic toxicity was leucopenia (7.7%) and the most common grade 3–4 non-hematologic toxicity was hypokalemia (11.5%).

In the postoperative setting with microscopically margin-negative resection, there are several concerns for further adjuvant therapies, such as residual diseases or not, performance status and tolerability of systemic regimens, and insurance coverage of novel target therapy or immunotherapy. In the CheckMate 577 trial [20], patients who had received nCRT with residual pathologic diseases were randomly assigned to adjuvant nivolumab or placebo for one year. Enrollment was regardless of programmed death receptor-1 ligand 1 (PD-L1) status. The median disease-free survival, the primary endpoint, was 22.4 months for the nivolumab group (95% confidence interval (CI), 16.6 to 34.0), as compared with 11.0 months (95% CI, 8.3 to 14.3) for the placebo group (hazard ratio for disease recurrence or death, 0.69; 96.4% CI, 0.56 to 0.86; *p* < 0.001). There were 29% of patients with SCC histology in the CheckMate 577 trial, and the median follow-up of 24.4 months was insufficient for overall survival. For patients without insurance coverage for nivolumab, the optimal approach is undefined, and administering adjuvant chemotherapy with a low-toxic regimen is a reasonable solution. The regimen with cisplatin and 5-FU in the present study resulted in grade 3–4 leucopenia in 19.1% of patients (previous literature 6% to 48.8%), and no grade 3–4 nausea/vomiting was noted (previous literature 2% to 11%). For the patients with intolerance to chemotherapy, novel systemic agents such as target therapy or immunotherapy may be a reasonable choice for adjuvant therapy. In the ATTRACTION-3 trial [21], nivolumab was associated with better survival than chemotherapy in patients with advanced esophageal squamous cell carcinoma refractory or intolerant to first-line chemotherapy. In the KEYNOTE-181 study [22], pembrolizumab as second-line therapy for advanced esophageal cancer did not improve overall survival in the whole population. There was a clinically meaningful improvement in OS with pembrolizumab versus chemotherapy in patients with SCC, but this was not statistically significant per prespecified boundaries. Patients with a programmed cell death ligand 1 (PD-L1) combined positive score (CPS) ≥ 10, which accounted for about 35% of the study population, had a median overall survival of 9.3 months with pembrolizumab versus 6.7 months with chemotherapy (HR 0.69; 95% CI 0.52–0.93; *p* = 0.0074). In the adjuvant setting after nCRT and surgery, further investigations are necessary to confirm the safety profile and survival impact of novel systemic agents in the prespecified population.

There were limitations in this study. This is a retrospective cohort study on a small sample from a single institution. However, its advantage is the homogenous treatment protocol of nCRT in all enrolled patients. Another advantage is the very high compliance of adjuvant chemotherapy (all patients in the adjuvant chemotherapy cohort completed two planned cycles of chemotherapy). Our results suggested that adjuvant chemotherapy likely improved OS in those with pN+ diseases, as shown by an apparent advantage with borderline significance in the subgroup analysis. More randomized trials are needed to confirm the role of adjuvant chemotherapy, and this study could be useful in the guidance for further designs of randomized trials in the future.

## 5. Conclusions

The most common failure pattern in the patients who received nCRT plus operation was distant failures, which account for nearly one-third of failure events. We analyzed patients who received adjuvant chemotherapy to discover the potential survival benefit in the patients with the residual nodal disease after nCRT plus operation. Patients with pN+ diseases had poor overall survival for locally advanced squamous cell carcinoma of the esophagus after nCRT plus operation, and adjuvant chemotherapy tended to improve overall survival in pN+ patients.

## Figures and Tables

**Figure 1 jpm-12-01252-f001:**
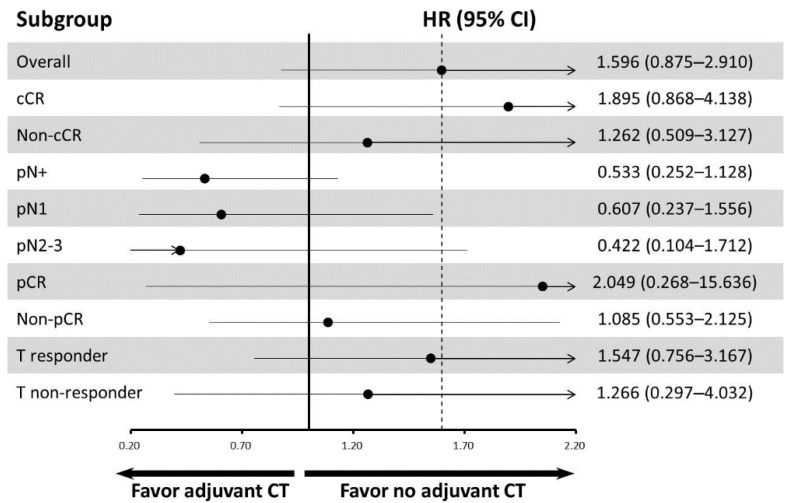
**Forest plot for subgroup analysis of overall survival.** Abbreviations: 95% CI, 95% confidence interval; adjuvant CT, adjuvant chemotherapy; cCR, clinical complete response; HR, hazard ratio; pCR, pathological complete response; pN+, pathologically node-positive; T responder, the patients with downstaging by T classification.

**Figure 2 jpm-12-01252-f002:**
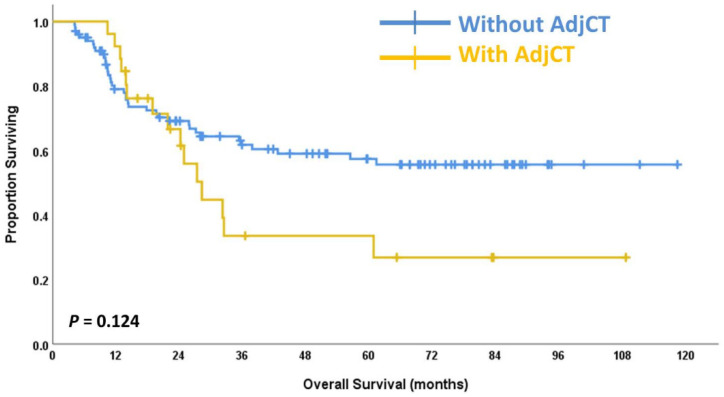
The overall survival curves for the group of patients receiving adjuvant chemotherapy (AdjCT) and the group without adjuvant chemotherapy (AdjCT).

**Figure 3 jpm-12-01252-f003:**
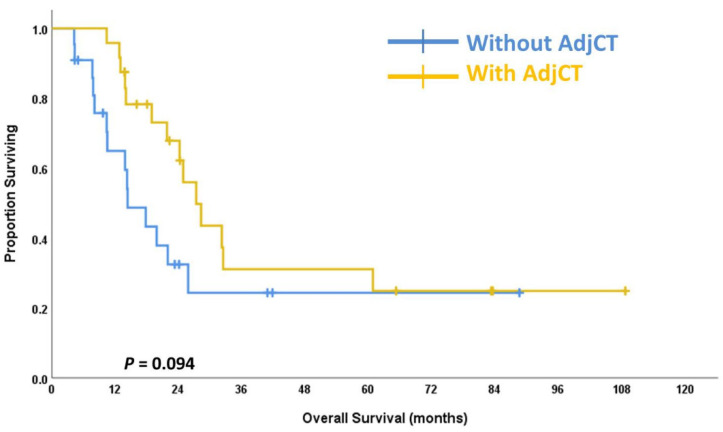
The overall survival curves for pathologic positive-node disease patients who received adjuvant chemotherapy (AdjCT) and patients who did not receive adjuvant chemotherapy (AdjCT).

**Table 1 jpm-12-01252-t001:** Patients’ characteristics.

		Total	SOC + Adjuvant CT	SOC + Follow-Up	*p*-Value
No. of Patients		127	26	101	
Sex	Male	122	23	99	0.058
	Female	5	3	2	
Age, years	Range	35–75	35–66	35–75	0.631
	Mean ± SD	54.77 ± 8.21	54.08 ± 8.63	54.95 ± 8.14	
Site	C	1	0	1	0.244
	U	15	2	13	
	M	24	2	22	
	L	80	19	61	
	2 sites	7	3	4	
cT	1	1	0	1	0.592
	2	3	0	3	
	3	120	26	94	
	4	3	0	3	
cN	0	6	1	5	0.909
	1	70	13	57	
	2	46	11	35	
	3	5	1	4	
pT	0	77	12	65	0.363
	1	13	4	9	
	2	17	4	13	
	3	20	6	14	
	4	0	0	0	
pN	0	81	2	79	<0.001
	1	32	18	14	
	2	12	5	7	
	3	2	1	1	
pCR		56	2	54	<0.001

Abbreviations: SOC, standard-of-care; SD, standard deviation; C, cervical; U, upper thoracic; M, middle thoracic; L, lower thoracic; cT, clinical T classification; cN, clinical N classification; pT, pathological T classification; pN, pathological N classification; pCR, pathological complete response.

**Table 2 jpm-12-01252-t002:** Acute toxicities during adjuvant chemotherapy.

Leucopenia	
Total	19 (73.1%)
Grade 1–2	17 (65.4%)
Grade 3–4	2 (7.7%)
Anemia	
Total	20 (76.9%)
Grade 1–2	19 (73.1%)
Grade 3–4	1 (3.8%)
Thrombocytopenia	
Total	9 (34.6%)
Grade 1–2	9 (34.6%)
Grade 3–4	0
Hypokalemia	
Total	7 (26.9%)
Grade 1–2	4 (15.4%)
Grade 3–4	3 (11.5%)
Hyponatremia	
Total	13 (50.0%)
Grade 1–2	10 (38.5%)
Grade 3–4	3 (11.5%)
Hypocalcemia	
Total	6 (23.1%)
Grade 1–2	4 (15.4%)
Grade 3–4	2 (7.7%)
Body weight loss	
Total	8 (41.9%)
Grade 1–2	8 (41.9%)
Grade 3–4	0

**Table 3 jpm-12-01252-t003:** Results of univariate and multivariate analyses with the Cox proportional hazards model on overall survivals.

		Univariate		Multivariate	
		Hazard Ratio (95% CI)	*p*-Value	Hazard Ratio (95% CI)	*p*-Value
Sex	(male–female)	0.415 (0.057–3.005)	0.384		
Age		1.008 (0.975–1.043)	0.632		
Histology	(grade 3–grade 1–2)	1.157 (0.661–2.024)	0.610		
Clinical stage			0.717		
III–II		1.510 (0.470–4.855)	0.489		
IVA–II		1.844 (0.413–8.245)	0.423		
Pre-Tx tumor length		1.034 (0.951–1.125)	0.430		
Time of CCRT to surgery		1.011 (0.995–1.026)	0.171		
Adjuvant CT (with–without)		1.596 (0.875–2.911)	0.128		
cCR (with–without)		0.456 (0.265–0.787)	0.005	0.527 (0.255–1.091)	0.085
pT			0.400		
pT1: pT0		1.281 (0.496–3.308)	0.609		
pT2: pT0		1.295 (0.549–2.827)	0.516		
pT3: pT0		1.819 (0.909–3.637)	0.091		
pN			<0.001		
pN1: pN0		2.320 (1.261–4.269)	0.007		
pN2: pN0		6.363 (2.853–14.194)	<0.001		
pN3: pN0		2.027 (0.271–15.152)	0.491		
pN+: pN0		2.838 (1.641–4.907)	<0.001	4.117 (1.366–12.404)	0.012
pCR (with–without)		0.466 (0.264–0.824)	0.009	1.883 (0.704–5.039)	0.207
pT size		1.264 (1.103–1.449)	0.001	1.188 (0.934–1.511)	0.160
No. of dissected LN		1.000 (0.982–1.018)	0.978		
LN ratio		1.024 (1.007–1.042)	0.007	1.005 (0.977–1.033)	0.736
T responder (SD–CR + PR)		1.886 (0.991–3.590)	0.053		
N responder			0.144		
SD–CR + PR		1.415 (0.779–2.569)	0.254		
PD–CR + PR		2.616 (0.916–7.467)	0.072		
Post-CCRT SUVmax		1.114 (0.978–1.269)	0.103		
ΔSUVmax/pre-CCRT SUVmax		0.984 (0.974–0.994)	0.003	0.988 (0.976–1.001)	0.066

Abbreviations: CI, confidence interval; Pre-Tx tumor length, pretreatment tumor length by endoscopy; CCRT, concurrent chemoradiotherapy; Adjuvant CT, adjuvant chemotherapy; cCR, clinical complete response; pN+, pathologically node-positive; pCR, pathological complete response; pT size, pathological tumor size; LN, lymph node; LN ratio, the ratio of positive lymph nodes over dissected lymph nodes; SD, stable disease; PR, partial response; PD, progressive disease; Post-CCRT SUVmax, maximum standardized uptake value of the esophageal tumor after concurrent chemoradiotherapy; ΔSUVmax, the change of maximum standardized uptake value of the esophageal tumor before and after concurrent chemoradiotherapy; pre-CCRT SUVmax, maximum standardized uptake value of the esophageal tumor before concurrent chemoradiotherapy.

## Data Availability

The datasets generated during and/or analyzed during the current study are not publicly available owing to confidentiality issues, but are available from the corresponding author upon reasonable request.

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
