# Peer review of "Adjuvant Chemotherapy for Pathological Node-Positive Disease in Squamous Cell Carcinoma of the Esophagus after Neoadjuvant Chemoradiotherapy Plus Surgery"

_jpm, 2022, doi:10.3390/jpm12081252_

Round 1

Reviewer 1 Report

I found this manuscript quite interesting. However, some deficiencies and discrepencies have been observed, that must be rectified.

1.     Introduction section is too brief. Authors should improve it.

2.     In the “Materials and Methods” of abstract, authors have mentioned that “days 1 through 4 and 22 through 25” were adopted for adjuvant chemotherapy. But in section 2.2., it is Day 1 to 4 (cycle 1) and Day 29 to Day 32 (cycle 2). Why this discrepancy?

3.     Authors should cross-check all the parameters. Different values for the parameters were given at different places. For example, in Section 3 (Results), median age is 56 years old, whereas in the Table 1, it is 54.77.

4.     Discussion part needs to be improved with addition of latest literature. Only article from 2021 and nothing from 2022.

5.     Further, conclusion was just of one sentence.

Reviewer 2 Report

In this manuscript, the authors analyzed the survival rate with adjuvant chemotherapy of the 127 locally advanced esophageal cancer patients. With the comparison of different methods in other related studies, the authors concluded that adjuvant chemotherapy can improve OS. Overall, the study is well-performed with clear designs and the results directly support the conclusions. The study can be important in improving our current understanding of the treatments in esophagus cancer patients. The manuscript is suggested to be accepted in the journal. 

Author Response

This manuscript is a resubmission of an earlier submission. The following is a list of the peer review reports and author responses from that submission.